# Bacteria Associated with the Roots of Common Bean (*Phaseolus vulgaris* L.) at Different Development Stages: Diversity and Plant Growth Promotion

**DOI:** 10.3390/microorganisms11010057

**Published:** 2022-12-24

**Authors:** Ricardo Rocha, Tiago Lopes, Cátia Fidalgo, Artur Alves, Paulo Cardoso, Etelvina Figueira

**Affiliations:** 1Department of Biology, University of Aveiro, 3810-193 Aveiro, Portugal; 2CESAM-Centre for Environmental and Marine Studies & Department of Biology, University of Aveiro, 3810-193 Aveiro, Portugal

**Keywords:** plant growth-promoting rhizobacteria, plant life cycle, growth-promoting traits, indole-3-acetic acid, siderophores, antifungal capacity, vegetative stage, flowering stage, pod stage

## Abstract

Current agricultural methodologies are vulnerable to erratic climate and are dependent on cost-intensive fertilization to ensure high yields. Sustainable practices should be pursued to ensure food security. *Phaseolus vulgaris* L. is one of the most produced legumes worldwide and may be an alternative to reduce the environmental impact of meat production as a reliable source of high-quality protein. Plant growth-promoting rhizobacteria (PGPR) are emerging as a sustainable option to increase agricultural production. To understand the dynamics between plants and microorganisms, the culturable microbiota of bean roots was isolated and identified at distinct stages of plant development (early and late vegetative growth, flowering, and pod) and root compartments (rhizoplane, endosphere, and nodules). Diversity and abundance of bacteria associated with root compartments differed throughout the plant life cycle. Bacterial plant growth promotion (PGP) and protection abilities (indole-3-acetic acid production, siderophore synthesis, and antifungal activity) were assessed and associated with plant phenology, demonstrating that among the bacteria associated with plant roots, several strains had an active role in the response to plant biological needs at each stage. Several strains stood out for their ability to display one or more PGP traits, being excellent candidates for efficient stage-specific biostimulants for application in precision agriculture.

## 1. Introduction

The use of chemical fertilizers, pesticides, and irrigation has enabled a steep increase in food production, sustaining the world population in the last few decades, but with high impacts on both human and natural systems. The current world population is around 7.7 billion, and by 2100, the forecast is for a 54% increase, which is about 11.88 billion people [1]. Population increases will impose new challenges on the already-established Sustainable Development Goals (SDGs) formulated by the United Nations (UN). To achieve these goals and meet the pressure for food production, sustainable approaches must be pursued, since the expected increase in food demand is approximately 58% to 70% [2], and sustainable approaches that increase agricultural production are important to achieve food security while protecting the environment.

The common bean (*Phaseolus vulgaris* L.) is one of the legumes most produced and consumed by humans, both dry and green, with an annual production in 2020 of 28 million tons (MT) and 23 MT, respectively [3]. Beans have a high nutritional value and are an important source of dietary protein for millions of people, providing essential amino acids, minerals (iron, copper, and zinc), vitamins (folate and pantothenate), and antioxidants (flavonoids) [4,5]. As the world population is constantly increasing, beans appear as a sustainable and low-cost alternative to meat.

The application of biostimulants based on plant growth-promoting bacteria (PGPB) is one of the possibilities to increase food security and food nutritional value, protecting the soil, maintaining the services it provides, and producing safer food for human consumption [6,7].

According to Zhang et al. [8], the microbiota associated with each species is an extension of its own genome, and it has been recognized that plant-associated microorganisms enhance plant growth, facilitate nutrient acquisition, and reduce different types of stress, including protection against several infectious pathogens [9]. This cooperation is so valuable to plants that approximately 5–30% of their photosynthates are made available for use by microorganisms [10]. Their ability to biologically fix nitrogen (N_2_) is considered to be one of the most crucial ecological services that microorganisms can provide to plants, thereby reducing the application of inorganic nitrogen fertilization [11]. Moreover, plant growth is also directly related to the ability of microbes to produce indole-3-acetic acid (IAA), siderophores, solubilize phosphate, and produce organic volatile compounds (VOCs) [12,13,14,15]. All of these growth-promoting characteristics mentioned can be found in PGPR. One of the most important is the production of IAA, which is one of the most physiologically active auxins (phytohormones) and plays a key role in different processes during plant development, such as increasing the length and number of roots, density of root hairs, enhancement of nutrient uptake, induction of flowering buds, and fruit setting [16,17,18,19]. Iron solubilization by siderophores is also crucial for plants, considering that iron in soil occurs mostly in non-bioavailable forms [20]. Hence, microorganisms that exhibit the ability to produce siderophores are essential during plant development because iron is a crucial micronutrient for several important biological functions, such as the regulation of protein activity, DNA synthesis, respiration, chlorophyll production, and as a component of several enzymes and electron-transfer proteins [21,22]. Indirect PGP traits such as antifungal activity are also essential for agricultural production. Plant fungal infections are one of the main causes of productivity losses in the agricultural sector [23]. For instance, microorganisms that can exhibit the capacity to restrain fungal infections are important, producing similar results to the application of chemical fungicides [24].

The significant role of microorganisms in agricultural productivity is an indication of their potential to support a more sustainable food production system. Hence, understanding the role of microbial communities deserves urgent attention, as it may provide information that can boost crop productivity.

The impetus for this study came from the need to verify if: (1) the interaction between soil cultivable bacteria and plant root changes throughout the plant life cycle, and (2) the plant growth-promoting abilities of the root-associated cultivable microbiota differ among different developmental stages. These dynamics may be related to specific plant needs at each stage of plant development, allowing the use of cultivable bacteria in precision agriculture and inducing the optimized development at each stage of the plant life cycle. The hypotheses were evaluated by assessing alterations in the cultivable biodiversity of the microbiota associated with different root compartments of bean plants at different growth stages and by evaluating the ability of isolated microorganisms to produce PGP traits at each developmental stage.

## 2. Materials and Methods

### 2.1. Plant Growth Conditions

Plants were grown in equivalent parts of soil and washed and autoclaved sand (*v*/*v*) to prevent alterations in the community of microorganisms present in the soil (40.538657 N, −8.693409 E). The addition of sand made the soil lighter, facilitating the collection of roots.

The mixture was used to fill black 15 L plastic pots with 10 L of the soil mixture. In each pot, five bean seeds (*Phaseolus vulgaris* L., variety “Patareco” NaturaSementes, Portugal, Lot Number 417, from 2020, and expiration date in July 2023) were sown in early July 2020. Plants were grown in a greenhouse environment and watered with sterile deionized water twice a day, in the morning and evening, to maintain constant soil moisture. The average recorded temperature varied between 18 and 22 °C, with daily thermal amplitudes between 5 and 8 °C.

The experimental design included four harvest periods, coinciding with bean plant development stages: V1—beginning of vegetative growth, one week after germination; V2—end of the vegetative growth, three weeks after germination; F—flowering stage with the flowers fully developed, five weeks after germination; and P—pod stage, seven weeks after germination, with pods of different sizes, some with fully developed beans but still green.

### 2.2. Plant Harvesting

At each plant developmental stage, three plants from different pots were randomly collected, and the shoots were detached from the roots. The root systems were shaken to release the aggregated soil trapped in the root system, maintaining the rhizoplane intact. Roots were kept on moist paper and transported to the laboratory under refrigerated conditions. From each plant root, two 3 cm apex samples and two nodules (larger and pinker) were selected.

### 2.3. Isolation of Bacteria

Bacteria were isolated following the method described by Somasegaran & Hoben [25] with modifications. Rhizoplane bacteria (Out) were isolated by dragging the root apex through the surface of yeast extract mannitol (YMA) plates containing 1 g/L mannitol [25]. To isolate endophytic bacteria (In), root apexes were first sterilized by soaking in 96% ethanol for 10 s, immersed for 2 min in a 30% hydrogen peroxide solution, and then rinsed twice in sterile deionized water. After that, apexes were cut transversally with a scalpel, and with the help of a sterilized toothpick, the inner parts of the root were collected and used to inoculate the YMA medium. To ensure no contamination of endophytic bacteria by outside bacteria, sterilized intact root apexes were dragged onto YMA before being cut. Bacteria from nodules (Nodules) were isolated from nodules, surface sterilized as described for root apexes, and crushed. Sterilized nodules were also dragged on the YMA before being crushed to ensure no contamination with bacteria present outside of the nodule. Macerate was streaked onto YMA plates. The plates were incubated in the dark for 10 days at 26 °C. Morphologically distinct single colonies were isolated by re-streaking. Following this methodology, 351 isolates were obtained during four stages of plant development.

#### Preservation of Bacterial Isolates

A loopful of each isolate was used to inoculate a 5 mL YMB tube. Tubes were incubated at 26 °C for 2–5 days, depending on the growth rate of the bacteria, and 500 μL of the resulting cultures were transferred to microtubes. To each microtube, 500 μL of sterile glycerol (30%) prepared in YMB medium was added, and the microtubes were vortexed and stored at −80 °C.

### 2.4. Bacteria Identification

#### 2.4.1. PCR-Based Fingerprinting

The 351 isolates were typed using BOX repetitive element-polymerase chain reaction (BOX-PCR) to screen for unique fingerprints before 16S rRNA gene amplification. Thus, duplicate genotypes can be identified, and only a representative isolate of each fingerprint pattern can be identified by 16S RNA gene sequencing. 

Isolates were inoculated on YMA, and single colonies were used to prepare a bacterial suspension in 50 μL autoclaved deionized water. Each PCR tube contained a mixture of 1 μL bacterial suspension and 1 μL BOXA1R primer (5′-CTACGGCAAGGCGACGCTGAC-3′) [26], as described by Cardoso et al. [27].

The PCR products were electrophoresed on an agarose gel (1.5%). Each gel (20 wells) contained two ladders (NZYDNA Ladder III, MB04401, NZYTech, Portugal), one negative control, and 17 samples. Electrophoresis was performed at 80 volts for 70 min. After staining the gel for 20 min in ethidium bromide, it was immersed in distilled water for 20 min to remove excess staining. The gels were then observed and scanned under ultraviolet light to obtain the individual typing profiles. GelCompar II (Applied Maths, Belgium) was used to calculate the Pearson correlation coefficient and analyze the clusters formed by applying the unweighted pair group method with arithmetic mean (UPGMA. A representative isolate from each fingerprint was randomly selected. The procedure yielded 227 representative isolates.

#### 2.4.2. 16S rRNA Gene Amplification and Phylogenetic Analysis

The 16S rRNA gene from the representative isolates was amplified using primers 27f (5′-AGAGTTTGATCCTGGC TCAG-3′) [28] and 1492r (5′-GGTTACCTTGTTACGACTT-3′) [28] and the NZYTaq 2× Green Master Mix (NZYTech, Portugal) in 25 μL microtubes by applying one cycle at 95 °C (7 min), 30 cycles at 94 °C (1 min), 53 °C (0.5 min), 72 °C (1.5 min), and a final cycle at 72 °C (10 min).

Agarose gels (1% agarose) were then electrophoresed. Each gel had 20 wells, two ladders, and one negative control, and the others were filled with samples. Electrophoresis was performed at 80 volts for 45 min, and the gel was stained for 20 min in ethidium bromide and then 20 min in distilled water to remove excess staining. The gels were then observed and scanned under ultraviolet light to obtain a clean bar at position 1400 bp in comparison with the NZYTech ladder control.

The PCR products were sent to Eurofins Genomics (Ebersberg, Germany) for sequencing. Each sequencing reaction tube contained the results of the PCR clean-up protocol or purification of DNA from enzymatic reactions using the NZYGelpure kit (NZYTech, Lisboa, Portugal). Sequences (between 546 and 1140) were edited using FinchTV V1.4.0 (Geospiza, USA). A BLAST search against the GenBank database was performed to identify bacteria at the genus level. Partial 16S rRNA gene sequences from the referred isolates were deposited in GenBank (ON419136-ON419292).

### 2.5. Plant Growth Promotion Abilities

Three plant growth-promoting (PGP) abilities were evaluated in the 227 isolates: siderophore production, indole-3-acetic acid (IAA) production, and antifungal capacity.

#### 2.5.1. Production of Siderophores

To assess the ability of bacteria to produce siderophores, strains were grown for ten days on YMA medium supplemented with chrome azurol S (CAS) solution consisting of 1.21 mg/mL CAS, 0.1 mM FeCl3·6H2O, and 1.82 mg/mL hexadecyltrimethylammonium bromide (HDTMA). The presence of an orange halo around colonies was considered positive for siderophore production [29]. Data are presented as the ratio of halo diameter to colony diameter.

#### 2.5.2. Production of Indole-3-Acetic Acid (IAA)

To quantify IAA production, strains were grown in 5 mL tubes of yeast-mannitol broth (YMB) supplemented with 100 μg/mL of tryptophan, the precursor of IAA. Strains were grown in the dark at 26 °C and 150 rpm, until 1.0 optical density at 600 nm or when maximum growth was reached after five days. After centrifugation for 10 min at 10,000× *g* and 4 °C, the supernatant was collected, and 500 μL of supernatant was reacted with 200 μL of Salkowski reagent (0.5 M FeCl_3_·6H_2_O, 35% HClO_4_) for 10 min at room temperature. The absorbance was measured at 530 nm using a spectrophotometer. IAA concentration was determined using IAA (Sigma) as a standard [30]. Data are presented as the concentration of IAA (μg/mL) normalized to the optical density (OD) of each strain.

#### 2.5.3. Screening Antifungal Capacity

Because bacteria exhibited slower growth than fungi, YMA plates were first inoculated with bacteria in the dark at 26 °C. The bacterial incubation time varied between 1 and 5 days depending on the bacterial isolate. When the colonies reached 0.5 cm, the center of the plate was inoculated with a 4 mm diameter plug from the peripheral zone of the fungal colony (*Fusarium oxysporum* L3W_8, GenBank Accession: OP071238) to guarantee that active hyphae were used. *F. oxysporum* was selected because it is recognized as one of the major causes of losses in agriculture worldwide in several different crops, including the common bean [31]. Plates were sealed with parafilm to ensure that if some of the bacterial antifungal compounds were volatile, they were retained in the plate environment [13]. Plates inoculated with fungus alone were used as controls. All plates were incubated until fungal colonies from the control plates reached a radius of 3 cm, and all isolates were examined in 3 replicates. At that time, the radius of the fungal colony growing with bacteria was measured, and growth inhibition was calculated ((C − B)/C × 100), where C is the fungal colony radius grown under control conditions (C), and B is the fungal colony radius grown in the presence of bacteria (Appendix A).

### 2.6. Statistical Analysis

All parameters tested were subjected to Monte Carlo tests with 9999 permutations using PRIMER 6 and PERMANOVA+ [32,33]. The null hypothesis was that, for each parameter, no significant differences existed between the growth stages and root compartments. Significant differences were considered only when *p* < 0.05, and were identified using different lowercase letters.

Data from the plant growth-promoting parameters were used to calculate the Euclidean distance similarity matrix for PGP traits. This similarity matrix was simplified by calculating the distance between centroids based on the conditions, and was then subjected to ordination analysis, performed by principal coordinate ordination (PCO). Pearson correlation vectors of plant growth-promoting parameters (correlation ≥ 0.97) were specified as supplementary variables on the PCO graph, allowing the identification of the descriptors that contributed most to the differences observed among the stages evaluated. 

Venn diagrams were constructed using a tool from Bioinformatics and Evolutionary Genomics, University of Ghent (http://bioinformatics.psb.ugent.be/webtools/Venn/ (accessed on 1 January 2022)), to compare the presence of bacterial genera throughout the different stages of bean plant development.

## 3. Results

### 3.1. Diversity

#### 3.1.1. Total Diversity

The identified diversity of the cultivable microbiota associated with bean roots at different developmental stages is shown in Figure 1a and Appendix A. Of the 351 isolates subjected to PCR-based fingerprinting, 227 distinct profiles were obtained. 16S rRNA gene-based identification and further characterization resulted in 201 strains belonging to 26 genera. The most represented genera were *Pseudomonas*, *Enterobacter*, *Bacillus*, and *Flavobacterium*, with 43, 26, 24, and 22 strains, respectively. *Priestia*, *Kosakonia*, *Variovorax*, *Delftia*, and *Achromobacter* were also well represented (14, 10, 8, 7, and 5, respectively). The four most common genera represented 61% of the identified diversity. The second group of the most represented genera accounted for 23.2% of the observed diversity. The least represented genera were *Chitinophaga* and *Microbacterium* with four strains; *Rhizobium* and *Leclercia* with three strains; *Curtobacterium*, *Stenotrophomonas*, and *Mucilaginibacter* with two strains; and *Flexibacter*, *Azospirillum*, *Trinickia*, *Cronobacter*, *Pantoea*, *Klebsiella*, *Acidovorax*, *Cupriavidus*, *Agrobacterium*, and *Paraburkholderia* with one strain each.

Diversity throughout the plant life cycle (Figure 1b and Appendix A) evidenced both differences and similarities. Five genera were common in all stages: *Achromobacter*, *Flavobacterium*, *Bacillus*, *Priestia*, and *Pseudomonas*. The genera *Enterobacter* and *Delftia* were recorded at all stages, except the pod stage (P). *Variovorax* was identified at all stages, except the late vegetative stage (V2). *Chitinophaga* were found at the flowering stage (F) and final vegetative stage (V2). *Rhizobium* was found at the pod (P) and late vegetative (V2) stages. Four genera were only present in the early vegetative stage (V1): *Pantoea*, *Leclercia*, *Stenotrophomonas*, and *Klebsiella*. Six genera (*Trinickia*, *Cronobacter*, *Curtobacterium*, *Flexibacter*, *Kosakonia*, and *Azospirillum*) were exclusive to the late vegetative stage (V2). Five genera appeared exclusively in the pod stage (P): *Paraburkholderia*, *Agrobacterium*, *Microbacterium*, *Acidovorax*, and *Cupriavidus*. No genera were found to be associated solely with the flowering stage (F).

#### 3.1.2. Diversity at Different Growth Stages

The diversity associated with different root compartments in plants at different growth stages is presented in Figure 2a and Appendix A. Twelve bacterial genera were identified at the beginning of vegetative growth (V1). The most abundant genera were *Enterobacter*, *Pseudomonas*, *Flavobacterium*, and *Bacillus*, accounting for 72.1% of the total, with *Enterobacter* being the most represented genus in the three root compartments. *Enterobacter* (29%), *Flavobacterium* (21%), *Pseudomonas* (14%), and *Bacillus* (14%) were the most abundant genera in the nodules (N). Inside the roots (In), *Enterobacter* (33%), *Pseudomonas* (17%), and *Delftia* (16%) were the most abundant. Outside the roots (Out), *Enterobacter* (29%), *Pseudomonas* (20%), and *Flavobacterium* (20%) were the most represented genera.

During late vegetative growth (V2), a higher diversity of bacterial genera (15 genera) was observed. *Pseudomonas*, *Kosakonia*, *Enterobacter*, and *Flavobacterium* were the most abundant genera, accounting for 55% of the total. Within this stage, the most represented genera were *Pseudomonas* (31%) and *Flavobacterium* (23%) for nodules, *Kosakonia* (27%), *Enterobacter* (23%), *Pseudomonas* (23%) inside the root, and *Enterobacter* (15%) outside the root. Higher diversity (12 genera) was observed outside the roots (9 genera) and nodules (6 genera).

At the flowering stage (F), three genera, *Pseudomonas*, *Bacillus*, and *Flavobacterium*, were the most abundant, totaling 67% of the strains. Within this stage, five genera were identified in the nodules, with *Pseudomonas* (38%) and *Achromobacter* (25%) being the most abundant; six genera inside the root, with *Bacillus*, *Priestia*, and *Flavobacterium* being the most abundant (totaling 66%); and seven genera outside the root, with *Pseudomonas* (38%), *Bacillus* (24%), and *Flavobacterium* (14%) being the most abundant.

Thirteen genera were identified in the pod stage (P). *Pseudomonas* and *Bacillus* were the most abundant (47% of the strains), followed by *Priestia*, *Variovorax*, and *Microbacterium* (33% of the strains). In the nodules, 50% of the strains belonged to the *Pseudomonas* genus. Inside the roots, three genera were more represented (*Pseudomonas*—26%, *Bacillus*—26%, and *Priestia*—22%). *Bacillus* (22%) and *Pseudomonas* (17%) were the most represented genera outside the root, but their diversity was higher, and two more genera were identified.

Analysis of each root compartment (Figure 2b and Appendix A) evidenced that *Pseudomonas* was common to all stages in the nodules. *Flavobacterium* was recorded at all stages, excluding the pod stage (P). *Enterobacter* was present only during the vegetative stage (V1 and V2). In contrast, the genus *Variovorax* was only recorded during the reproductive stages (F and P stages). In the nodules, 11 genera exclusive to one stage were found. *Enterobacter* was present at all stages inside the root (In), except for the P stage. *Achromobacter* and *Pseudomonas* were recorded in all stages, but not in the F stage. The genus *Variovorax* was identified in all stages, but not in the V2 stage. *Bacillus* was found in all stages, but not in V1. The genus *Delftia* was found at Stages V1 and V2. *Chitinophaga* and *Flavobacterium* were recorded in stages F and V2, respectively. The genus *Priestia* was observed in stages F and P. Two genera were exclusively found inside the root at V1, two other genera (*Flexibacter* and *Kosakonia*) at V2, and *Microbacterium* and *Cupriavidus* were specific to P. Outside the root (Out) *Flavobacterium*, *Bacillus*, *Priestia*, and *Pseudomonas* were present at all stages. *Enterobacter* was recorded in all stages, but not in the P stage. The genus *Delftia* was identified in Stages V2 and F. Fourteen genera were identified in only one stage, especially V2 and P.

### 3.2. Plant Growth Promotion

#### 3.2.1. IAA Production

The IAA production ability of the strains differed with plant development (Figure 3a). Significant differences were observed between the beginning of vegetative growth (V1) and flowering (F), with the highest mean value observed for V2. Differences among the root compartments at each stage were also noticed (Figure 3b). At the beginning of vegetative growth (V1), nodules and rhizoplane (Out) bacteria produced lower, although not statistically different, IAA levels than bacteria from inside the root (In). During late vegetative growth (V2), similar IAA levels were produced among bacteria from the three root compartments. At the flowering stage (F), differences among root compartments were also not significant, but rhizoplane bacteria (Out) showed a lower ability to produce IAA than endophytic bacteria (nodules and roots). At the pod stage (P), differences were not significant, but nodule bacteria displayed lower production, and rhizoplane (Out) displayed higher IAA production.

#### 3.2.2. Siderophores

Differences among developmental stages (Figure 4a) showed significant differences between V1 and P stages, but no significant differences in V2 and F stages were observed. Among root compartments (Figure 4b) no significant differences were observed for each developmental stage among the development stages. However, in the late vegetative stage (V2), bacteria from nodules showed a higher ability to produce siderophores than bacteria from the rhizoplane and inside the root. In the pod stage, the opposite was observed, with bacteria from nodules displaying a lower ability to produce siderophores than exophytic and inside the root bacteria.

#### 3.2.3. Antifungal Capacity

The overall analysis of antifungal capacity (Figure 5a) showed that a high proportion of strains had weak or no ability to inhibit *Fusarium oxysporum* growth. However, inhibition higher than 50% was registered at all stages, especially at the pod stage, with a significantly higher antifungal activity than in the first three growth stages. 

A more detailed analysis highlighted the differences among the different root compartments for each stage (Figure 5b). At the beginning of vegetative growth (V1), no significant differences among root compartments were observed (bar charts), but the distribution by level of inhibition varied (circle charts), with the number of isolates with higher antifungal activity (>20%) being lower in bacteria from nodules and higher in those from the rhizoplane. At the end of vegetative growth (V2), significant differences were observed between bacteria isolated from the outside (Out) and inside roots (In) (bar charts). The distribution by the level of inhibition (circle charts) showed that the lower antifungal activity of rhizoplane bacteria resulted from the vast majority (34 out of 36) of the isolates having no or a reduced capacity (<20%) to inhibit fungal growth. The higher antifungal capacity of the bacteria inside the root was due to the number of isolates (6 out of 37) with a strong capacity to inhibit the fungus. At the flowering stage (F), bacteria from the root inside (In) had the lowest antifungal capacity, and bacteria from the rhizoplane (Out) had the highest antifungal capacity. At the pod stage (P), no significant differences were observed among the root compartments (bar charts). The distribution by the level of inhibition (circle charts) was also similar, despite the difference in the number of isolates from each root compartment.

### 3.3. Principal Components Ordination (PCO)

PCO1 appeared as the main axis (69.7%), explaining the variation in the PGP ability of bacteria isolated from the roots of bean plants at different growth stages (Figure 6). Along PCO1, the pod stage (P) was positioned on the positive side of the axis, flowering stage (F) next to the origin, and vegetative stages (V1 and V2) on the negative side of Axis 1. PCO2 accounted for 27.7% of the total variation, with the flowering stage (F) on the positive side, early vegetative stage (V1) next to the origin of the axis, and late vegetative (V2) and pod (P) stages on the negative side of Axis 2. From PCO analysis, it was possible to observe that the production of siderophores and antifungal capacity were strongly correlated with the P stage, and IAA production was strongly correlated with vegetative growth, especially in the late vegetative stage (V2). 

## 4. Discussion

The bacterial communities associated with plant roots can be influenced by several factors [34]. In this study, the cultivable bacterial community associated with the bean root system was identified and found to change throughout the plant life cycle, and the changes observed appeared to meet the specific needs of the plant at each developmental stage (Appendix A).

The first hypothesis, that the diversity of cultivable bacteria associated with roots differs with plant development, which has already been hypothesized by other authors [35,36], was confirmed by our data (Figure 7). *Pseudomonas*, *Bacillus*, *Priestia*, *Variovorax*, *Enterobacter*, *Kosakonia*, and *Flavobacterium* were the most common genera in the cultivable root microbiota of bean plants, and changes throughout the life cycle were noted. The genus *Enterobacter* was the most represented during the initial vegetative stage (V1). The genera *Pseudomonas*, *Flavobacterium*, and *Bacillus* were present throughout the plant life cycle, although differences in representativity among the stages were observed. *Pseudomonas* and *Bacillus* were more abundant in the reproductive stages (F and P). *Flavobacterium* was less abundant in the pod stage (P). In the vegetative stage, *Kosakonia* increased from V1 to V2. Indeed, the *Kosakonia* genus participates in some processes specific to this stage of development, such as root branching and growth, nutrient uptake, and water uptake, thus favoring vegetative plant development [37,38,39]. 

In the reproductive stages (F and P), the genus *Priestia* appeared to be highly represented in relation to the vegetative stages (V1 and V2). *Variovorax* was also highly represented in the pod stage. In contrast to the other three stages, the genus *Enterobacter* was not present at the pod stage (P). The apparent loss of *Enterobacter* during late bean plant development can be linked to alterations in roots at this stage [40], possibly due to an unfavorable environment for *Enterobacter* growth. In contrast, *Bacillus* increased steadily during the flowering and pod stages (F and P). According to Chaparro et al. [35], it is possible that a basal microbiota exists in bean plants, as evidenced by the genera present at all developmental stages, including *Pseudomonas*, *Flavobacterium*, *Bacillus*, and *Priestia*. Although there were fewer isolates, several genera of bacteria were common in all four developmental stages. This basal microbiota may be complemented by other bacterial genera that are specific or more abundant at certain stages of plant development. The literature shows the capabilities of several strains of these poorly represented and sporadic genera to exhibit growth-promoting characteristics that may participate in plant development directly, through IAA or siderophore production, or indirectly, through protection against phytopathogenic fungi [41,42,43,44].

As an insight into the diversity in root compartments, the cultivable bacterial community present in nodules was much more specific and conserved throughout the plant life cycle, possibly indicating the particular conditions present in nodules where only some bacteria are able to grow. Inside the root, differences were observed throughout plant development, evidencing that conditions inside the root changed, initially supporting (V1) the growth of *Enterobacter*, *Pseudomonas*, *Delftia*, and some other genera less represented. At V2, *Delftia* was replaced by *Kosakonia*. At F and P, *Bacillus* and *Flavobacterium*, which are minor genera, became dominant, and the genus *Priestia* also increased. Thus, in bacteria colonizing the root, major changes were observed between the plant vegetative (V1 and V2 stages) and reproductive (F and P stages) growth, possibly evidencing different growth requirements, more related to the development of root and photosynthetic organs initially and changing to flowering and seed growth later [45]. Outside the root, differences among stages were also observed, with *Leclercia*, *Stenotrophomonas*, *Trinickia*, *Kosakonia*, *Curtobacterium*, *Azospirillum*, *Cronobacter*, and *Chitinophaga* being exclusive to vegetative growth (V1 and V2) and *Paraburkholderia*, *Rhizobium*, *Agrobacterium*, *Microbacterium*, and *Variovorax* being exclusive to the pod stage.

At the beginning of development (V1), the bacteria associated with the root may reflect the diversity present in the soil, and no major changes in diversity were observed within the three root compartments. Over time, the bacterial community may adapt to the different conditions present inside and outside the roots, favoring the genera most adapted to the prevalent conditions in each compartment, and the microbiota became more diverse among compartments [46]. These variations may also have followed the changes in the conditions present in root tissues caused by plant development, initially linked to the development of vegetative organs, providing more resources for root development, and later privileging the mobilization of photosynthates to the production of seeds [45].

The second hypothesis, that changes in diversity would reflect alterations in the PGP ability of strains meeting plant needs at crucial moments of its development, was also confirmed by our data (Figure 7).

Antifungal activity was similar until the flowering stage (V1, V2, and F) but increased significantly at the pod stage, validating the hypothesis that PGPR capacity varies throughout the plant life cycle. Differences among root compartments were also observed, with higher activity occurring inside the root (V2), outside (F), or not differing (V1 and P). In the early vegetative stage (V1), the root system is actively growing, and associations with microorganisms are being established [47]. Thus, disputes over a place inside the roots may occur outside, with bacteria eliminating fungal competition. Indeed, higher antifungal activity was observed outside the root compared to the nodules and inside the root in V1. At the late vegetative stage (V2), both bacteria and fungi that successfully colonize the root system compete for photosynthates, with bacteria trying to slacken fungal growth. Our results may corroborate this assumption since the V2 antifungal activity inside the root is 3-fold higher than that in nodules and 6-fold higher than that outside. Bacteria isolated from the pod stage (P) had the highest antifungal capacity. It is possible that plants in this stage are more susceptible to fungal infection due to the forthcoming end of the plant’s life cycle, when plant organs start to senesce and become more prone to infection, such as the root nodules that initiate senescence immediately after flowering. During pod development, plants become less dependent on soil nutrients and more dependent on internal nutrient relocation from older vegetative organs to new reproductive organs. This decrease in root function leads to senescence. Higher antifungal activity may hinder the establishment of fungal infection in senescing organs, as observed in the results obtained, preventing the spread of fungal infection to other parts of the plant, such as pods, and enabling plants to complete their life cycle. Among the genera isolated, *Pseudomonas*, *Enterobacter*, *Bacillus*, *Priestia*, *Achromobacter*, *Microbacterium*, *Curtobacterium*, and *Trinickia* showed antifungal activity, confirming the information presented in the literature [43,48,49,50,51,52,53]. Additionally, the obtained results highlighted *Flavobacterium* as a genus with high antifungal ability, with some strains inhibiting fungal growth above 60%.

IAA production was significantly different between the early vegetative (V1) and flowering (F) stages, but the results showed that the highest capacity was detected in isolates from the late vegetative (V2) and pod (P) stages, validating the hypothesis that PGPR capacity varies throughout the life cycle. Auxins are a class of phytohormones that regulate the entire period of plant growth and development in processes such as cell division, elongation, and differentiation [54]. According to Wu et al. [54], IAA synthesis in the roots was higher at the end of plant vegetative growth, which is in agreement with the results obtained. High levels of IAA before flowering have been documented in several studies and have been associated with the role of IAA in flowering induction, indicating that bacterial PGP abilities are linked with plant needs at specific developmental stages. The same authors reported that IAA production was lower during the flowering stage, which corroborates these results. This study showed several genera to have the capacity to produce IAA, such as *Priestia*, *Variovorax*, *Kosakonia*, *Leclercia*, *Pseudomonas*, *Enterobacter*, *Bacillus*, *Flavobacterium*, *Microbacterium*, *Rhizobium*, *Leclercia*, *Mucilaginibacter*, and *Cupriavidus*. In addition to its use as a biostimulant, the capacity of the bacterial strains tested to synthesize IAA can be taken into account for the production of IAA, replacing the synthetic alternative and making the process more efficient and sustainable [55]. Some genera have already been reported as being able to produce IAA [56,57,58,59,60,61,62,63,64,65].

Siderophore production may reflect the plant’s need for iron. In the late vegetative stage (V2), the bacteria in the nodules showed a high ability to synthesize siderophores. At this stage, the most effective nodules (higher and pinker) were observed. The effectiveness (and pink color) of nodules is linked with leghaemoglobin, a red-colored protein containing iron, responsible for controlling oxygen diffusion and protecting the oxygen-sensitive enzymes from denaturation, such as nitrogenase in bacteroids, but at the same time supplementing sufficient oxygen for bacterial respiration [66]. Nitrogenase is an enzyme present in all nitrogen-fixing bacteria that is responsible for nitrogen fixation and is a metalloenzyme with Fe-S clusters essential for its activity. Thus, at the late vegetative stage (V2), nodules require high amounts of iron, and the siderophore-producing bacteria associated with the nodule can help attain this [67]. Knowing that beans are rich in iron and, as expected, a higher ability of bacteria to produce siderophores was observed in the pod stage, this PGP ability may indicate the contribution of bacteria to the plant to meet its iron needs at this developmental stage. Furthermore, iron is a cofactor in the synthesis of hormones such as ethylene, which is produced mainly at the end of the annual plant life cycle [68]. In this study, strains belonging to *Chitinophaga*, *Flexibacter*, and *Trinickia* showed the ability to produce siderophores, adding new genera to the list that already included *Pseudomonas*, *Enterobacter*, *Bacillus*, *Flavobacterium*, *Kosakonia*, *Variovorax*, *Delftia*, *Priestia*, *Achromobacter*, *Microbacterium*, *Rhizobium*, *Mucilaginibacter*, *Azospirillum*, *Acidovorax*, *Cupriavidus*, *Agrobacterium*, *Paraburkholderia*, and *Curtobacterium* [38,69,70,71,72,73,74,75].

## 5. Conclusions

The results from this study indicate a high cultivable diversity of genera associated with bean roots, which changes throughout plant development. Some genera were common throughout the plant life cycle, while others were specific to one or more developmental stages, and sporadic genera were also encountered.

Higher bacterial specificity in the internal tissues was observed both in the nodules and in the internal parts of roots, indicating more constant and defined conditions inside the plant. The more diverse conditions outside the root certainly received the plant’s influence through root exudates but also from external pressures, supporting the higher bacterial diversity observed.

Bacteria play important roles in the regulation and development of the plant life cycle. The variation in the observed diversity of bacteria associated with the bean root was reflected in the PGP ability of the strains, possibly meeting the plant needs at vital moments of its development, which was evidenced by the differences observed in bacterial PGP traits associated with each developmental stage.

At each plant stage, several strains stood out for their ability to produce one or more PGP traits. Tailoring inoculants, not only to specific developmental stages but also to specific conditions and crops, can become a new option to produce biostimulants that are better adapted to plant-specific needs. Thus, propelling precision agriculture will contribute to more sustainable and environmentally friendly food production.

## Figures and Tables

**Figure 1 microorganisms-11-00057-f001:**
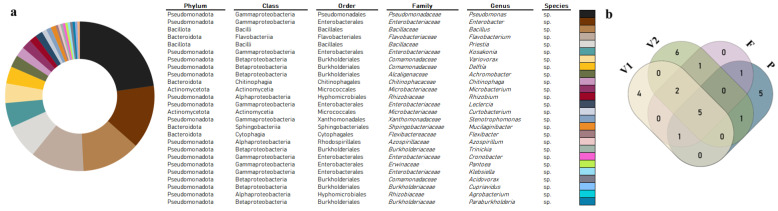
Global distribution of bacterial genera. (**a**) Circle chart showing global bacterial distribution by genera found in *Phaseolus vulgaris* L. during plant development. (**b**) Venn diagram of bacteria genera distribution in the four stages of plant development: beginning of vegetative stage—V1, end of vegetative stage—V2, flowering stage—F, and pod stage—P.

**Figure 2 microorganisms-11-00057-f002:**
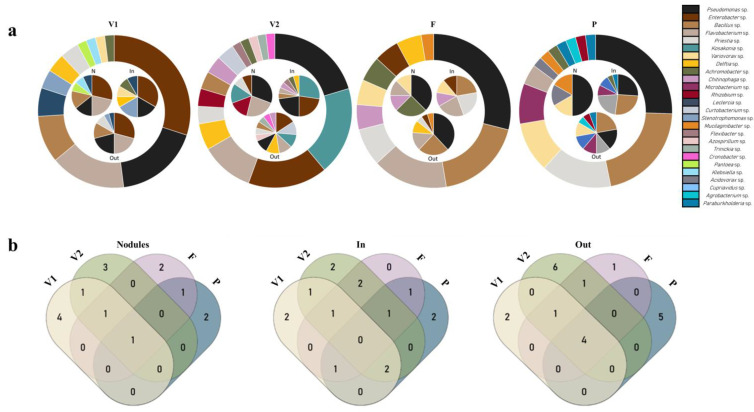
Specific distribution of bacterial genera. (**a**) Circle charts showing the distribution of bacterial genera according to plant development stage (outer circle) and root compartment (inner circles). Plant development stages: beginning of the vegetative stage (V1), end of the vegetative stage (V2), flowering stage (F), and pod stage (P). Root compartments: nodules (N), inside roots (In), and outside roots (Out). (**b**) Venn diagrams of bacteria genera distribution by root compartments (Nodules, In, Out) at four stages of plant development (V1, V2, F, P).

**Figure 3 microorganisms-11-00057-f003:**
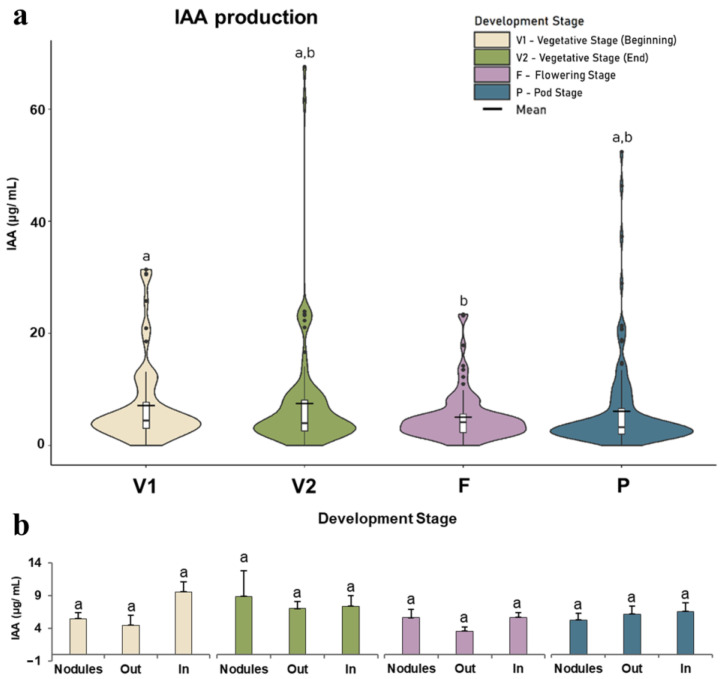
Bacterial ability to produce indole-3-acetic acid (IAA). (**a**) Violin plot representation of the overall IAA production ability of bacteria isolated at each plant growth stage (beginning of the vegetative stage, V1; end of the vegetative stage, V2; flowering stage, F; and pod stage, P). The outlier values (black dots) and averages (black lines) are marked. (**b**) Bar chart representation of the average IAA production in each root compartment (rhizoplane outside the root—out, inside the root—in, and in nodules—nodules). Between 65 strains at stage V1 and 115 strains from stage P were used to construct violin plots. Chart bar values are the means ± standard deviation of IAA production from 15 strains belonging to the V1 stage from outside (out-of-root compartment) to 50 strains from the P stage in Out (root compartment). Significant differences (*p* < 0.05) among developmental stages (violin plots) and root compartments (bar charts) are represented by different lowercase letters (a and b).

**Figure 4 microorganisms-11-00057-f004:**
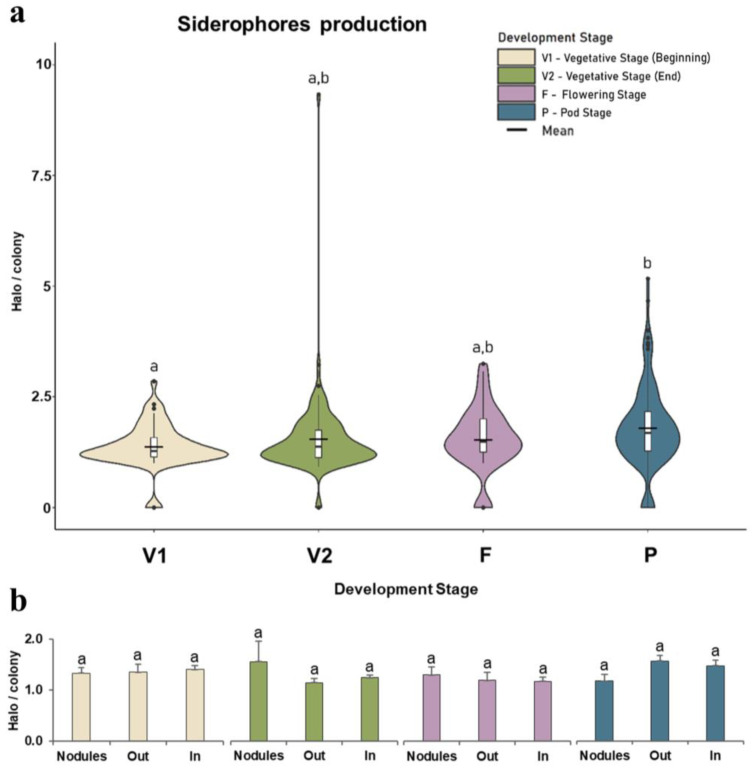
Bacterial ability to produce siderophores. (**a**) Violin plot representation of the overall siderophore production ability of bacteria isolated at each plant growth stage (beginning of the vegetative stage, V1; end of the vegetative stage, V2; flowering stage, F; and pod stage, P). Outlier values (back dots) and average (black line) are marked. (**b**) Bar chart representation of the average siderophore production in each root compartment (rhizoplane outside the root—out, inside the root—in, and in nodules—nodules). Sixty strains at stage V1 and 108 strains from the P stage were used to construct violin plots. Chart bar values are the means ± standard deviation of siderophore production from 15 strains belonging to the V1 stage from Out and Nodules (root compartments) to 49 strains from the P stage in Out (root compartment). Significant differences (*p* < 0.05) among development stages (violin plots) and root compartments (bar charts) were represented by different lowercase letters (a and b).

**Figure 5 microorganisms-11-00057-f005:**
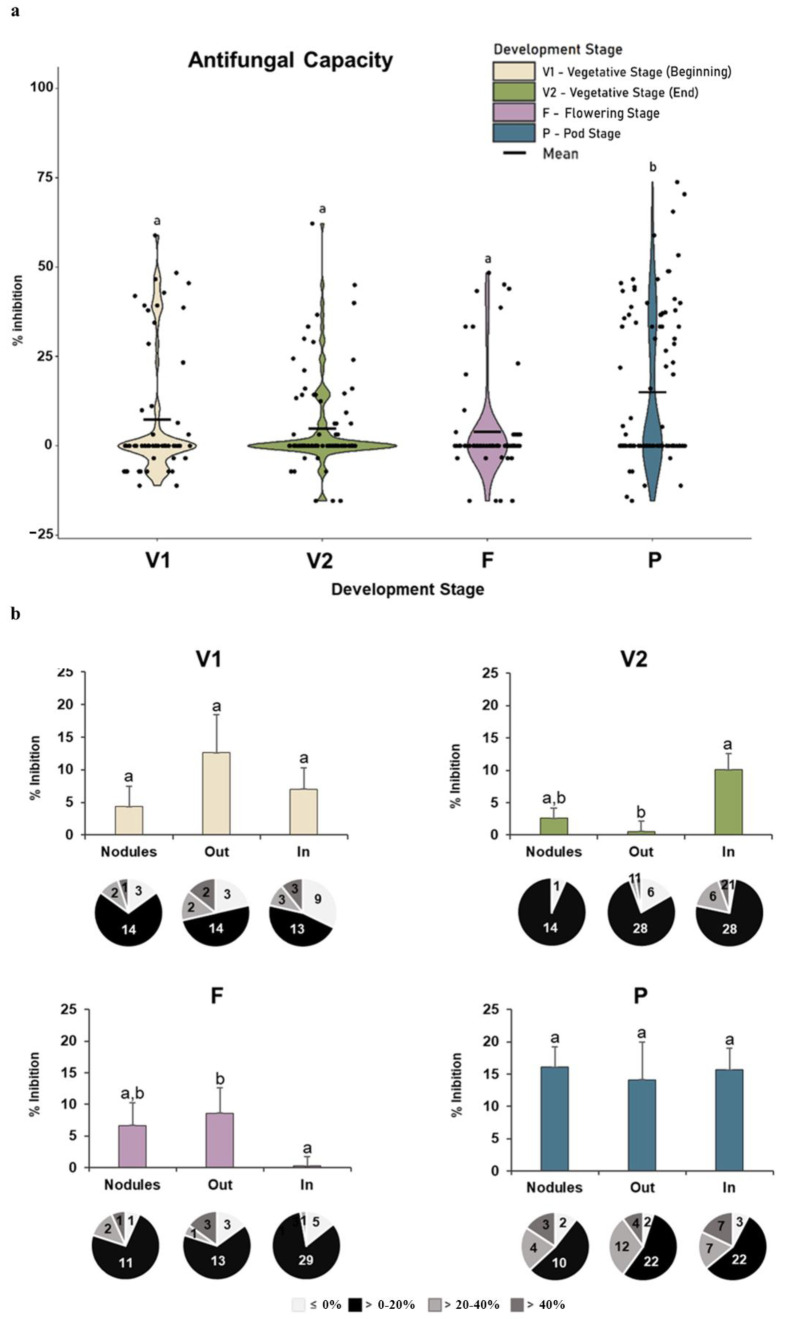
Antifungal capacity of *Fusarium oxysporum*. (**a**) Violin plot representation of the overall antifungal capacity of the bacteria isolated at each plant growth stage (beginning of vegetative stage, V1; end of vegetative stage, V2; flowering stage, F; and pod stage, P). The values of inhibition (black dots) and average values (black line) are marked. (**b**) Average (bar charts) and quantity of strains per level of inhibition (circle charts) of antifungal activity (% inhibition of fungal growth) in each root compartment (rhizoplane outside the root—out, inside the root—in, and in nodules—nodules). The 62 strains at stage V1 and 98 strains from the P stage were used to construct violin plots. Chart bar values are the means ± standard deviation of antifungal activity from 14 strains belonging to the V1 stage outside the root (Out) to 40 strains from the P stage in Out. Significant differences (*p* < 0.05) among development stages (violin plots) and root compartments (bar charts) were represented by different lowercase letters (a and b).

**Figure 6 microorganisms-11-00057-f006:**
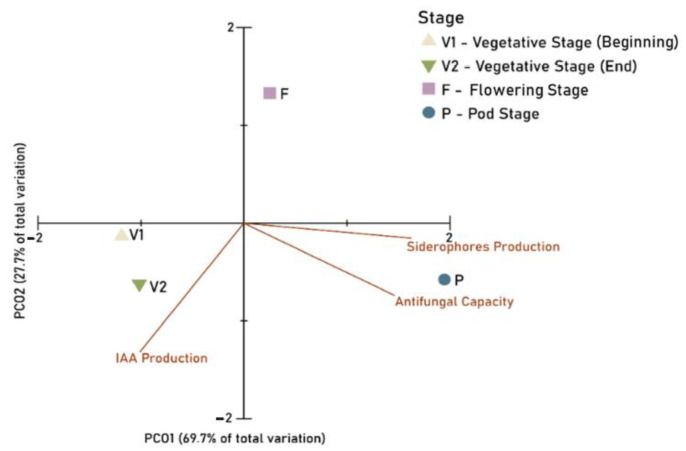
Principal coordinate ordination (PCO) of plant growth-promoting abilities at different stages of plant development (V1, V2, F, P). PGP traits were then superimposed.

**Figure 7 microorganisms-11-00057-f007:**
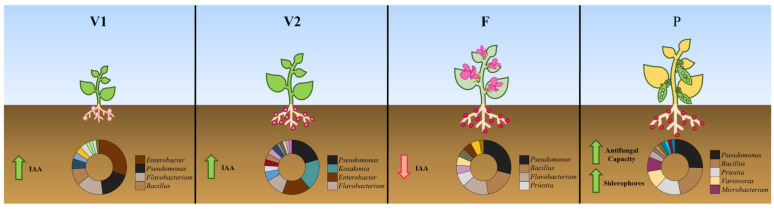
Overview of bacterial genera distribution and major impacts of bacterial diversity at different stages of *Phaseolus vulgaris* L. development. V1: beginning of vegetative growth (7 days after emergence), V2: end of vegetative growth (21 days after emergence), F: flowering stage (35 days after emergence), P: P-pod stage (49 days after emergence). The main plant growth-promoting abilities are as follows: IAA, production of 3-indole acetic acid, siderophore production, and antifungal activity towards *F. oxysporum* (green arrows indicate high ability; red arrows pointing down represent low ability). Bacterial diversity at the genus level is expressed in a circular chart, with the name of the most included.

## Data Availability

Raw data are available on request.

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
