# Peer review of "Bacteria Associated with the Roots of Common Bean (Phaseolus vulgaris L.) at Different Development Stages: Diversity and Plant Growth Promotion"

_microorganisms, 2022, doi:10.3390/microorganisms11010057_

Round 1

Reviewer 1 Report

The manuscript is written well. The data is well presented.

Author Response

The authors would like to thank the reviewer for all comments.

Reviewer 2 Report

Dear Corresponding Author
I checked your paper and have some comments to you to improve your review paper:
For the works such as your work it is better using metagenomics/metabarcoding method to evaluate the bacteria associated with a plant root in different growth stages. After receiving data as FASTQ files you may use SEED2 program (a Win-based software) to analyze and extract the final results.
When you want to present your data as Figure 1 you need to have different stages as Phylum, Order, Family, genus and species.
All of scientific names should be as italics (For example: line 282-288). You will need to correct all of them throughout the text.
When you speak about antifungal activity you will need to add some plate-photos to show the ability as in vitro.
With Best Regards
Reviewer

Author Response

The authors would like to thank the reviewer for all comments and suggestions that greatly improved the quality and clarity of the manuscript. All comments were taken into consideration and the manuscript was revised accordingly.

All changes are tracked by Microsoft Word tracker

For the works such as your work it is better using metagenomics/metabarcoding method to evaluate the bacteria associated with a plant root in different growth stages. After receiving data as FASTQ files you may use SEED2 program (a Win-based software) to analyze and extract the final results.

Response: The authors agree with the suggestion, and believe it would give another interesting perspective to the work presented, but since the objective of the work was to isolate bacterial strains with the potential to be tested as biostimulants, the focus of the work was mainly on culturable strains, since they offer the possibility to be kept under laboratory conditions and can be easily grown to be applied as biostimulants .

When you want to present your data as Figure 1 you need to have different stages as Phylum, Order, Family, genus and species.

Response: The figure1 was changed to adress the suggestion of the reviewer

All of scientific names should be as italics (For example: line 282-288). You will need to correct all of them throughout the text.

Response: For some techincal reason in the manuscript submited  the names were italicized but were lost. Authors have revised the document to ensure that species names are now italicized.

When you speak about antifungal activity you will need to add some plate-photos to show the ability as in vitro.

Response: Photos of antifungal activity were added in the manuscript as a supplementary figure S2.

Reviewer 3 Report

Dear authors,

I have reviewed your manuscript " Bacteria associated to the roots of Phaseolus vulgaris L. at different development stages: diversity and plant growth promotion " submitted for publication in Microorganisms. The manuscript is interesting and presents a valuable collection of information on the abundance of bacteria associated with bean roots in different stages of plant development and their ability to produce indole-3-acetic acid, siderophore, and antifungal activity.

I have highlighted a few points below that could improve the quality of the manuscript before publication.

A genus (or genus group) is always italicized. Please check this throughout the document.

Line 2: Consider adding " common bean (Phaseolus vulgaris L)" in the title.

Lines 41-45: Long sentence. Consider splitting it into two sentences.

Line 50: According to Zhang et al. [8]

Line 54: 5–30%

Line 57: nitrogen fertilization [add a reference to support this sentence].

Line 100: 18–22 °C

Line 101: 5–8 °C

Line 103: V1 – beginning

Lines 116-117: Somasegaran & Hoben [24]

Line 115: Was there a specific amount of roots and nodules?

Line 119: Please include a reference to this medium.

Line 129: Please standardize the units (26°C or 26 °C). Earlier you added a space between the values and the symbol.

Line 129: Were they incubated in light or dark?

Line 140: Describe “BOX-PCR”

Line 147: Cardoso et al. [26].

Line 183: It is necessary to correct the units. You used a / earlier instead of -1; please standardize this throughout the document.

Line 189: Cells? Please double check this.

Line 191: Dark conditions?

Line 197: add temperature and light conditions used during inoculation.

Line 205: bean [30]. Plates…

Lines 237-238: This should be in italics. Pseudomonas, Enterobacter, Bacillus, and Flavobacterium.

Lines 238-246: Same observation as lines 237-238. A genus (or genus group) is always italicized. Please check this throughout the document.

Line 365: It would be great if you add pictures of the plates.

Line 447: Chaparro et al. [34]

Line 517: Wu et al. [53]

Line 526: Add some practical applications of these bacteria with IAA production capacity. For example, use as an auxin source in micropropagation protocols. This reference (https://cpsjournal.org/2022/02/17/cps2022003/) supports my observation.

Line 638: Is this correct? Please double check this reference.

Author Response

The authors would like to thank the reviewer for all comments and suggestions that greatly improved the quality and clarity of the manuscript. All comments were taken into consideration and the manuscript revised accordingly.

All changes are tracked by Microsoft Word tracker

A genus (or genus group) is always italicized. Please check this throughout the document.

Response: The manuscript originally have the names italicized, with the exception of Enterobacter (line 307). For some reason in the submission process names in italics desapeared. The problem is now is corrected.

Line 2: Consider adding " common bean (Phaseolus vulgaris L)" in the title.

Response: Added.

Lines 41-45: Long sentence. Consider splitting it into two sentences.

Response: Splitted in two sentences.

Line 50: According to Zhang et al. [8]

Response: Corrected.

Line 54: 5–30%

Response: Corrected.

Line 57: nitrogen fertilization [add a reference to support this sentence].

Response: Added.

Line 100: 18–22 °C

Response: corrected

Line 101: 5–8 °C

Response: Corrected.

Line 103: V1 – beginning

Response: Corrected.

Lines 116-117: Somasegaran & Hoben [24]

Response: Corrected.

Line 115: Was there a specific amount of roots and nodules?

Response: Yes, there was a specific number of roots and nodules. The structure of the sentence was reformulated in the manuscript to be more clear.

Line 119: Please include a reference to this medium.

Response: Added.

Line 129: Please standardize the units (26°C or 26 °C). Earlier you added a space between the values and the symbol.

Response: Standardized.

Line 129: Were they incubated in light or dark?

Response: The plates were incubated in dark conditions. This is now refered in the text.

Line 140: Describe “BOX-PCR”

Response: Added.

Line 147: Cardoso et al. [26].

Response: Corrected.

Line 183: It is necessary to correct the units. You used a / earlier instead of -1; please standardize this throughout the document.

Response: Standardized.Thank your for noticing. A revision was made throughout the text and all the units are now in “/”

Line 189: Cells? Please double check this.

Response: Corrected.

Line 191: Dark conditions?

Response: Corrected (dark condition).

Line 197: add temperature and light conditions used during inoculation.

Response: Added.

Line 205: bean [30]. Plates…

Response: Corrected.

Lines 237-238: This should be in italics. Pseudomonas, Enterobacter, Bacillus, and Flavobacterium.

Response: For some techincal reason in the manuscript submited the names were italicized but were lost. Authors have revised the document to ensure that species names are now italicized.

Lines 238-246: Same observation as lines 237-238. A genus (or genus group) is always italicized. Please check this throughout the document.

Response: Corrected.

Line 365: It would be great if you add pictures of the plates.

Response: Added as a supplmentary figure S2.

Line 447: Chaparro et al. [34]

Response: Corrected.

Line 517: Wu et al. [53]

Response: Corrected.

Line 526: Add some practical applications of these bacteria with IAA production capacity. For example, use as an auxin source in micropropagation protocols. This reference (https://cpsjournal.org/2022/02/17/cps2022003/) supports my observation.

Response: Added. We thank the reviewer for the improved discussion on the IAA, which makes it easier to understand the importance of the topic.

Line 638: Is this correct? Please double check this reference.

Response: No, it was not correct. Revised and changed

Round 2

Reviewer 2 Report

Dear authors,

Thanks for addressing all my comments. I can recommend your manuscript for publication.

Best regards